# Bending Strength of Continuous Fiber-Reinforced (CFR) Polyamide-Based Composite Additively Manufactured through Material Extrusion

**DOI:** 10.3390/ma17122937

**Published:** 2024-06-15

**Authors:** Maciej Łakomy, Janusz Kluczyński, Bartłomiej Sarzyński, Katarzyna Jasik, Ireneusz Szachogłuchowicz, Jakub Łuszczek

**Affiliations:** Institute of Robots & Machine Design, Faculty of Mechanical Engineering, Military University of Technology, Gen. S. Kaliskiego St., 00-908 Warsaw, Poland; maciej.lakomy@student.wat.edu.pl (M.Ł.); bartlomiej.sarzynski@wat.edu.pl (B.S.); katarzyna.jasik@wat.edu.pl (K.J.); ireneusz.szachogluchowicz@wat.edu.pl (I.S.); jakub.luszczek@wat.edu.pl (J.Ł.)

**Keywords:** carbon, three-point tests, Onyx, FDM, FFF, MEX, DIC, flexural strength, fracture analysis

## Abstract

This paper shows the three-point bending strength analysis of a composite material consisting of polyamide doped with chopped carbon fiber and reinforced with continuous carbon fiber produced by means of the material extrusion (MEX) additive manufacturing technique. For a comparison, two types of specimens were produced: unreinforced and continuous fiber-reinforced (CFR) with the use of carbon fiber. The specimens were fabricated in two orientations that assure the highest strength properties. Strength analysis was supplemented by additional digital image correlation (DIC) analysis that allowed for the identification of regions with maximum strain within the specimens. The utilization of an optical microscope enabled a fractographic examination of the fracture surfaces of the specimens. The results of this study demonstrated a beneficial effect of continuous carbon fiber reinforcement on both the stiffness and strength of the material, with an increase in flexural strength from 77.34 MPa for the unreinforced composite to 147.03 MPa for the composite reinforced with continuous carbon fiber.

## 1. Introduction

In recent years, additive manufacturing (AM) techniques, commonly known as 3D printing, have gained significant popularity. This is evidenced by the substantial number of articles related to the analysis of the potential applications of AM in the production of machinery and equipment components [1,2,3,4,5,6,7,8]. Still, there are a lot of potential applications of AM technologies such as temporary fastening [9,10,11], but due to too little knowledge about the behavior of materials, only conventional materials are being used. Despite the parallel development and research into the use of metals, additive techniques predominantly rely on polymers [12,13,14,15,16,17]. One such polymer is polyamide (PA) and the composite materials produced from it. Among the wide range of available polymers for additive manufacturing, polyamides hold a special place due to their unique properties and applications.

Polyamides, also known as nylons, are a group of thermoplastic polymers characterized by high mechanical strength, flexibility, and chemical resistance [18,19,20]. Additively manufactured parts made from PA exhibit high surface quality, enabling their direct application without the need for post-processing [21]. Leveraging these advantages, PA is used to produce components such as gears, handles, sliding elements, and machinery accessories [22,23].

AM techniques allow for the production of PA parts via Selective Laser Sintering (SLS), Multi Jet Fusion (MJF), and material extrusion (MEX) techniques [24]. In the case of SLS and MJF, the material is in powder form. In SLS, it is selectively sintered using a laser beam. During MJF manufacturing, the material is bonded using additional binders and sintering lamps [25]. The most commonly used techniques for manufacturing PA products are FFF/FDM (Fused Filament Fabrication/Fused Deposition Modeling) techniques which are commercial types of MEX [26]. Taking into account the nature and operating conditions of devices that produce parts using MEX techniques, the production process is straightforward and cost-effective [27]. In FDM/FFF, polyamide filaments are heated to melting temperature and then precisely deposited layer by layer, allowing for the creation of geometrically complex parts with high surface quality. Consequently, polyamides are employed in the production of prototypes, tools, and even end-use parts across various industries [28].

The appropriate selection of AM parameters allows for the obtaining of parts with favorable strength properties and relevant dimensional accuracy [29]. Before any material is introduced into the production of parts that are subjected to significant loads, it undergoes tests to evaluate its mechanical properties under various conditions. One of the key aspects of research on materials produced by AM is the study of flexural strength, which plays a crucial role in many engineering applications [30,31,32].

A significant advantage of polyamides is their ability to combine with other materials, which opens up possibilities for creating composites with enhanced mechanical properties [33,34,35]. For example, the addition of glass or carbon fibers to a polyamide matrix can significantly increase the stiffness and tensile strength of the components produced [36,37]. Based on an analysis of trends in new materials used in the production of parts via MEX additive techniques, there is notable interest in polyamide-based composites. Among them are the following:Glass fiber-reinforced polyamide (PA-GF)—glass fibers improve the tensile strength, stiffness, and dimensional accuracy of printed parts [38,39].Carbon fiber-reinforced polyamide (PA-CF)—carbon fibers added to polyamide increase mechanical strength and stiffness while reducing the weight of printed elements [40,41].Aramid fiber-reinforced polyamide (PA–Aramid)—aramid fibers (e.g., Kevlar) used as reinforcement improve the wear resistance, impact resistance, and temperature stability of printed elements [42,43].Graphite fiber-reinforced polyamide (PA-G)—graphite fibers used as reinforcement improve the compressive strength, chemical resistance, and thermal resistance of printed parts [44,45].

The MEX-CFR additive manufacturing technique, involving the use of two filament sources (a matrix of thermoplastic material and a reinforcement of continuous fiber, most often carbon), is currently under development [46,47,48]. Due to the limited number of available scientific articles addressing this topic, the authors of this manuscript undertook an attempt to conduct such research. This allowed for an analysis of the influence of print direction on the strength of the samples and the degree of reinforcement using continuous carbon fiber. The conducted studies aim to provide insight into the potential of MEX-CFR technology for producing high-strength polyamide-based composites reinforced with continuous carbon fiber and to contribute to the further development and optimization of this method for manufacturing advanced composite materials.

The conducted research on MEX-CFR material has made significant contributions to addressing gaps in this field. Previously, there has been a limited amount of data available on the properties and performance of carbon fiber-reinforced polyamide composites. This study provides new insights into the mechanical and thermal properties of MEX-CFR composites, filling a critical knowledge gap. The findings highlight the material’s improved fatigue strength, wear resistance, and chemical resistance, which were not well documented before. Additionally, this research offers detailed analysis on the processing challenges and solutions associated with MEX-CFR, providing valuable information for future studies. The enhanced damping properties and maintained surface aesthetics observed in this study further expand the understanding of MEX-CFR composites’ applications. Overall, this research sets a foundation for more comprehensive studies and practical applications of MEX-CFR materials in various industries.

## 2. Materials and Methods

To better illustrate the steps taken, Figure 1 presents a flow chart of the conducted research.

For the fabrication of the test samples, a Mark Two printer from Markforged (Waltham, MA, USA) was used, as presented in Figure 2. The dimensions of the machine building volume are 320 × 132 × 154 mm. The printer allows for the production of elements with a layer thickness ranging from 100 to 250 μm. This device is equipped with a dual-material extrusion head, which enables the even feeding of two materials—one in the form of a thermoplastic filament serving as the matrix of the produced element and the other in the form of a continuous fiber serving as reinforcement. This method is referred to as MEX-CFR (material extrusion with Continuous Fiber Reinforcement).

The test samples were manufactured using Markforged 800cc Onyx composite material, which served as the matrix for the produced elements, and continuous carbon fiber (CF-BA-50), also provided by Markforged, which served as the core (reinforcement) of the produced samples. The materials used, in the form of spools dedicated to the printer, are shown in Figure 3. To reduce the susceptibility of Onyx material to absorb water particles from the environment, it was stored in a dedicated, airtight case during the manufacturing process.

Onyx is a polyamide (nylon)-based composite reinforced with micro-carbon fibers. As a material intended for AM, it can be used alone or with various reinforcements (e.g., carbon fiber, Kevlar, or Fiberglass). Continuous fiber is utilized as internal layers within the fabricated elements (it cannot be used independently). Carbon fiber has the highest strength-to-weight ratio among all continuous fibers offered by Markforged. It is commonly used in components as a replacement for some aluminum alloys. Table 1 presents the selected mechanical properties of nylon, Onyx, and continuous carbon fiber.

The samples had dimensions of 120 × 15 × 6 mm, in accordance with the PN-EN ISO 178 standard [49]. Figure 4 illustrates the two orientations of the samples during the additive manufacturing process. To analyze the effect of continuous carbon fiber reinforcement on Onyx material, 5 samples were produced for each of the four variants:Five samples oriented along their larger side (along the “X” axis) without reinforcement (Figure 4a);Five samples oriented along their larger side (along the “X” axis) with reinforcement (Figure 4a);Five samples oriented along their smaller side (along the “Z” axis) without reinforcement (Figure 4b);Five samples oriented along their smaller side (along the “Z” axis) with reinforcement (Figure 4b).

During the design phase of the test samples, the placement of continuous carbon fibers serving as reinforcement was considered in the sample’s structure. Each sample contained four reinforcement layers, each 0.5 mm thick. These layers were spaced every 0.5 mm along the height of the sample. To better illustrate their placement, the dedicated software view for the Markforged printer system is shown in Figure 5. In this figure, the gray color represents the Onyx material, while the blue indicates the continuous carbon fiber reinforcement. The infill density of the sample was 100%.

After appropriately preparing the process, the production of the test samples commenced. A characteristic feature of this printer is the unheated bed. The nozzle temperature for the Onyx filament was set to 280 °C, while for the carbon fiber, it was set to 260 °C. The layer thickness during printing was 0.125 mm. The printer executed 5 external perimeters. The selected printing parameters are presented in Table 2. Figure 6 shows the Markforged Mark Two device and a close-up view of the print head during the additive manufacturing process of the test samples.

The manufacturing of the four series of samples took place in four separate processes. Figure 7 shows the view of five unreinforced samples manufactured along the X-axis. The next step involved preparing the samples for strength testing and deformation analysis using the DIC system.

The bending strength tests were conducted using an Instron 8802 servo-hydraulic testing system (Norwood, MA, USA). The DIC system from Dantec Dynamics (Ulm, Germany) was additionally employed to measure the deformations occurring during the strength tests. To identify the material structure, the samples were appropriately marked by applying individual paint spots. The tests were carried out following the PN-EN ISO 178:2019-06 standard [49] “Plastics–Determination of flexural properties”. The methods described in the standard utilize a three-point bending system. During the test, each sample rested on two supports on its wider side and was loaded with a rod placed at an equal distance from both supports. The applied force acted on the sample until its outer surface fractured or until a deformation of 5% was reached, whichever occurred first. The deformation rate used was 0.01 mm/min. Figure 8 shows the testing machine used and a frame captured during the bending test along with the DIC analysis.

To characterize the material’s crack propagation during three-point bending tests, fractographic analysis was conducted. These observations were carried out using the KEYENCE VHX-7000 device (Keyence, Osaka, Japan). 

## 3. Results and Discussion

### 3.1. Flexural Properties—Bending Test

The use of sensors and the testing machine’s software during the strength tests allowed for the registration and determination of key material properties. The force acting on the samples generated bending stresses. The displacement sensor enabled the determination of maximum sample deformations at the moment of maximum force application. Based on the obtained data, the Young’s modulus was calculated for each sample. The results are summarized in Table 3.

Based on the obtained results, calculations of the average values of individual material properties were performed for each type of sample. The results are presented in Table 4. The analysis of the obtained results shows that the unreinforced sample printed with its largest side on the XY plane achieved the lowest bending strength value of 71.47 MPa. The deformation of 4.46% may indicate the low stiffness of the Onyx material compared to samples reinforced with continuous carbon fiber. The sample printed along the “Z” axis without reinforcement achieved a maximum bending stress of 133.59 MPa with a deformation of 4.36%. Comparing stress levels for Onyx material samples, a significant influence of the sample orientation during printing on material strength can be observed. The results of bending tests for the carbon fiber-reinforced composite differ. In the case of samples printed along the “X” axis, the maximum bending strength value was 159.55 MPa, while reinforced samples printed along the “Z” axis reached a value of 148.73 MPa. Furthermore, the “X CFR” samples exhibited the highest stiffness during bending tests (the material deformation at which the maximum stress was reached was 2.38%). Using the obtained values, the Young’s modulus was calculated for each sample. The highest Young’s modulus value was achieved by the “X CFR” sample, while the lowest was observed for the “X” sample. The average values of individual parameters were calculated for each type of sample. Table 4 demonstrates the influence of print orientation and CFR reinforcement on the strength properties of individual samples.

Upon analyzing the results of the bending tests, it can be observed that for samples printed along the “Z” axis, the reinforcement with continuous carbon fiber (CCF) resulted in an increase in bending strength value by only about 11% (from 128.10 MPa to 143.38 MPa). However, the situation is different for samples printed along the “X” axis. In this case, the reinforcement with continuous carbon fiber led to a significant increase in the bending strength values of the samples from 77.34 MPa to 147.03 MPa. This represents a 90% increase in material bending strength. The addition of continuous carbon fiber reinforcement increased the stiffness of the material. For the “X” samples, stiffness doubled (by 92%). This stiffness is associated with the deformation that the sample undergoes when subjected to the maximum force. For the “X” samples, deformation decreased from 4.38% to 2.27%. For the “Z” samples, stiffness increased by just under 10% (deformation decreased from 4.34% to 3.95%). The CCF reinforcement had a significant impact on the calculated Young’s modulus values. For the “X” samples, the reinforcement caused a 150% increase in the Young’s modulus value (from 3.56 GPa to 8.74 GPa). However, for the “Z” samples, this increase was less significant, at just under 9%.

To better present and compare the obtained results, a graph was created showing the relationship between bending stress (σ_fM_) and sample deformation (ε). Representative samples were selected from each group based on achieving the highest bending stress values. The graph is displayed in Figure 9. The orange lines represent the bending behavior of samples produced along the “X” axis, while the blue lines represent samples produced along the “Z” axis. The dashed line indicates a sample made of Onyx composite, while the solid line represents a sample containing continuous carbon fiber reinforcement (CCF). From the analysis, it is evident that the “X CFR” sample exhibits the highest bending strength. The sample reached a maximum stress level of approximately 160 MPa with a deformation of around 2.5%. The weakest sample is “X”, which reached a maximum stress level of about 80 MPa with a simultaneous deformation of 4.5%.

### 3.2. Digital Image Correlation

For the analysis of material displacements during the bending test, the digital image correlation (DIC) method was employed. Figure 10 illustrates the measurement results for the maximum deformation of each sample.

For better comparison, views of the samples before the tests were conducted are presented. A color scale corresponding to the levels of sample deformation is provided along with all the images. In the first four images, the samples are shown before applying any load. As per the legend, the coloration of the sample corresponds to zero deformation. Changes occur when a force is applied to the sample. When the sample reaches maximum deformation, the characteristic behavior of the extreme upper and lower layers of the sample during the bending test is clearly visible. The upper layers are compressed, while the lower layers are stretched. Between the samples without and those with reinforcement, subtle differences in deformation distribution and hence stress distribution can be observed. In the case of samples without reinforcement, deformations spread more evenly across the entire sample, confirming its lower stiffness. In reinforced samples, deformations are more concentrated in one area. The orange (sometimes red) color corresponds to areas with the greatest deformations. Thanks to the DIC technique employed during the strength tests, it is possible to record the most stressed areas in the samples.

### 3.3. Fracture Analysis

Using a computerized microscope with digital image analysis, photographs of the sample fractures and continuous carbon fiber in cross-sections were captured. The results are presented in Figure 11. Figure 11a–d depict photographs of individual samples taken at a magnification of ×100. Figure 11e shows a single carbon fiber used as reinforcement at a magnification of ×800. During the imaging process, the samples were oriented in the direction consistent with the application of successive layers of material. In the images of the fractures of samples reinforced with continuous carbon fiber presented in Figure 11c,d, areas with a denser structure and a different fracture pattern are visible in their central part. This indicates the presence of reinforcement in the form of continuous carbon fiber.

To analyze the composition of the material, a higher magnification of the microscope lens was used. Figure 12 and Figure 13 present images of samples taken at magnifications of ×200 and ×400, respectively. Figure 12 focuses on analyzing the composition of samples printed with Onyx material without reinforcement. Based on this, reinforcement particles in the form of small fibers were observed, identified as chopped carbon fiber (indicated by red arrows).

Figure 13 shows images of samples that contain reinforcement in the form of continuous carbon fiber. There is a slight difference in appearance between continuous carbon fiber and chopped carbon fiber. Longer structures, identified as continuous carbon fiber, are noticeable. For a better analysis of the reinforcement morphology, a picture of a single carbon fiber was taken at a magnification of ×800. The result is shown in Figure 14.

Figure 14 shows a view of a single continuous carbon fiber at a magnification of ×800. Red arrows highlight areas where clusters of chopped material, similar to those observed in the Onyx composite, were noticed. Based on this, it was concluded that the continuous carbon fiber supplied by Markforged is nothing more than chopped carbon fiber embedded in polyamide. The difference lies in the content of chopped fiber in both materials—in the case of continuous carbon fiber (CCF), there is more of it compared to the Onyx material. The polyamide-based matrix allows for the free melting of the material and its deposition in both cases—both for the Onyx composite and the reinforcement.

Carbon fibers are compatible with polyamides in composites for several reasons. Carbon fibers have surfaces that bond well with polyamides, which is crucial for transferring mechanical loads between the fibers and the polymer matrix. Good adhesion ensures higher composite strength. Moreover, polyamides have chemical groups that can form hydrogen bonds or other interactions with the surface of carbon fibers, enhancing the compatibility of both materials. In addition, polyamides such as PA6 (polyamide 6) and PA66 (polyamide 66) have suitable processing temperature ranges that are compatible with carbon fibers. This allows for efficient composite processing without degrading the fibers. Polyamides are known for their good mechanical properties, such as strength and abrasion resistance. Combined with the high strength and stiffness of carbon fibers, they create composites with excellent mechanical properties. Polyamides can be processed using various techniques such as injection molding, forming, and extrusion, which are compatible with the processes used to produce composites with carbon fibers.

## 4. Conclusions

The conducted studies enabled the determination of the bending strength of specimens made of Onyx composite material both without reinforcement and reinforced with continuous carbon fiber. Through the utilization of DIC technique, the areas of greatest deformation in the specimens were identified. The analysis of the structure of the specimens allowed for the identification of continuous carbon fibers acting as reinforcement. Here are the main conclusions drawn from the conducted research:Utilizing the Markforged Mark Two device allowed for the production of specimens reinforced with continuous carbon fiber.Static bending tests revealed that reinforcement with continuous carbon fiber results in increased material strength. However, the direction of printing and application of subsequent material layers play a significant role. In the case of printing along the X-axis, reinforcement led to a significant increase in bending strength from 77.34 MPa to 147.03 MPa, achieving the highest value among all tested configurations. For prints along the Z-axis, the bending strength increased from 128.10 MPa to 143.38 MPa. Conducting five tests allowed us to obtain repeatable results.The use of DIC allowed for the identification of areas in the specimens that experienced the greatest deformation during bending tests.A microscopic analysis of fractures allowed for the identification of components in the material structure, both in the Onyx composite and continuous carbon fiber.Continuous carbon fiber (CCF) remains a composite material very similar to the Onyx composite. Both materials essentially consist of chopped carbon fiber particles embedded in polyamide (nylon). The difference lies in the content of chopped fiber in both materials. In the case of CCF, it is several times higher than in Onyx material, resulting in a significant increase in the strength and stiffness of specimens containing continuous carbon fiber reinforcement.The MEX-CFR (material extrusion with Carbon Fiber Reinforcement) technique has its specific advantages and disadvantages, both in terms of the process and the resulting composite of carbon fiber-reinforced polyamide. Compared to the simple MEX technique, it offers significant benefits such as increased mechanical strength, high stiffness, and a lower coefficient of thermal expansion. However, the drawbacks of this technique include higher material costs, process complexity, and shorter tool lifespan (e.g., extrusion nozzles) due to the abrasive properties of carbon fibers.The application of carbon fiber reinforcement in polyamide-based composites not only enhances flexural strength and stiffness but also significantly affects many other material properties. The fiber improves fatigue strength, wear resistance, and chemical resistance and reduces weight, enhances damping properties, and maintains high surface aesthetics.The results of the MEX-CFR research have potential applications in high-performance industries such as aerospace, automotive, and sports equipment, where enhanced mechanical properties and reduced weight are crucial. These findings can also be applied in the development of durable and lightweight consumer products. Future research should focus on optimizing the processing techniques to further improve the material’s performance and reduce manufacturing costs. Additionally, investigating the long-term environmental impact and recyclability of MEX-CFR composites will be essential for sustainable development.

## Figures and Tables

**Figure 1 materials-17-02937-f001:**
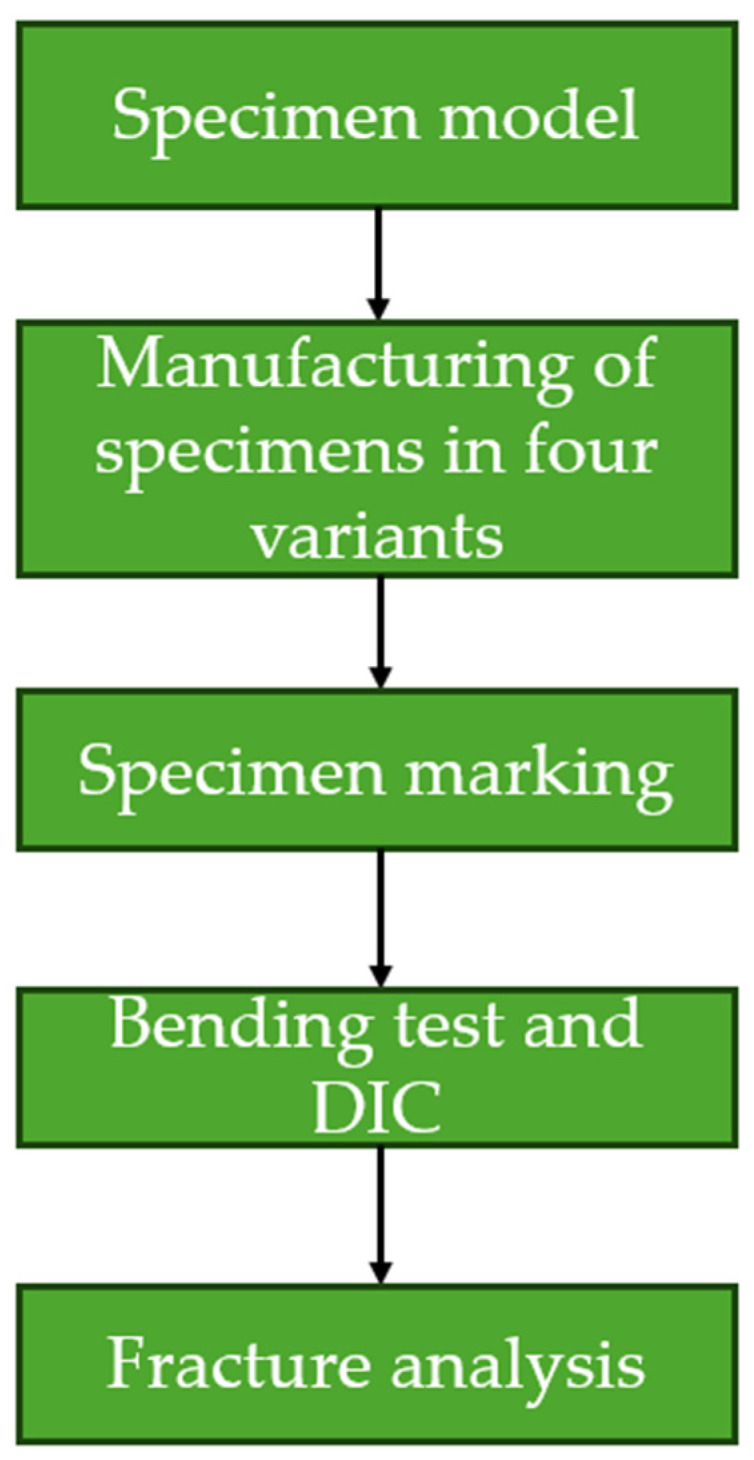
A flow chart showing the steps taken during research.

**Figure 2 materials-17-02937-f002:**
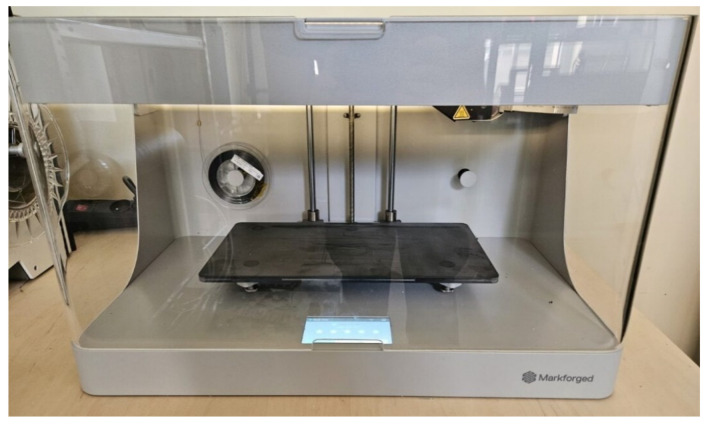
Markforged Mark Two printer used for manufacturing test samples.

**Figure 3 materials-17-02937-f003:**
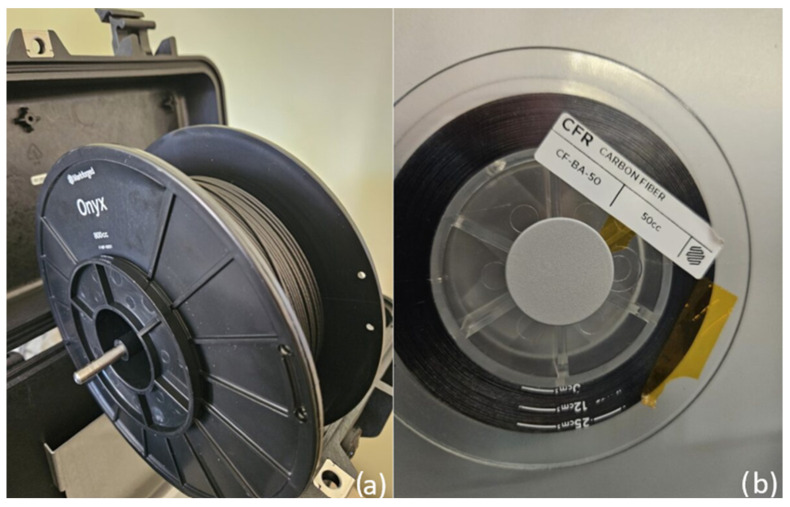
Materials used for manufacturing samples, (**a**) Onyx composite material, (**b**) continuous carbon fiber.

**Figure 4 materials-17-02937-f004:**
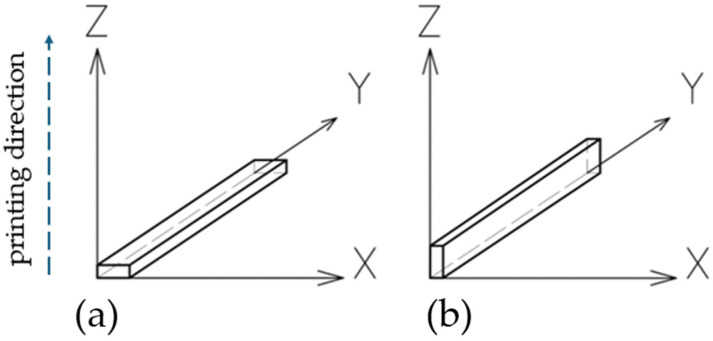
Orientation of test samples during MEX process: (**a**) along “X” axis, (**b**) along “Z” axis.

**Figure 5 materials-17-02937-f005:**
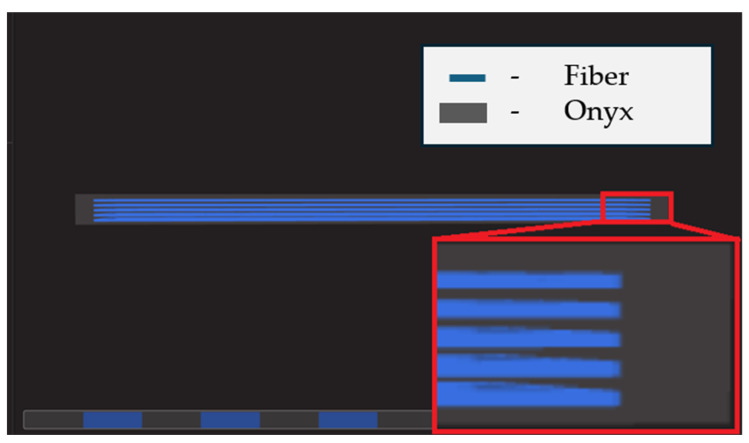
A view of the test sample during preparation for printing, highlighting the zones reinforced with continuous carbon fiber in blue, is shown.

**Figure 6 materials-17-02937-f006:**
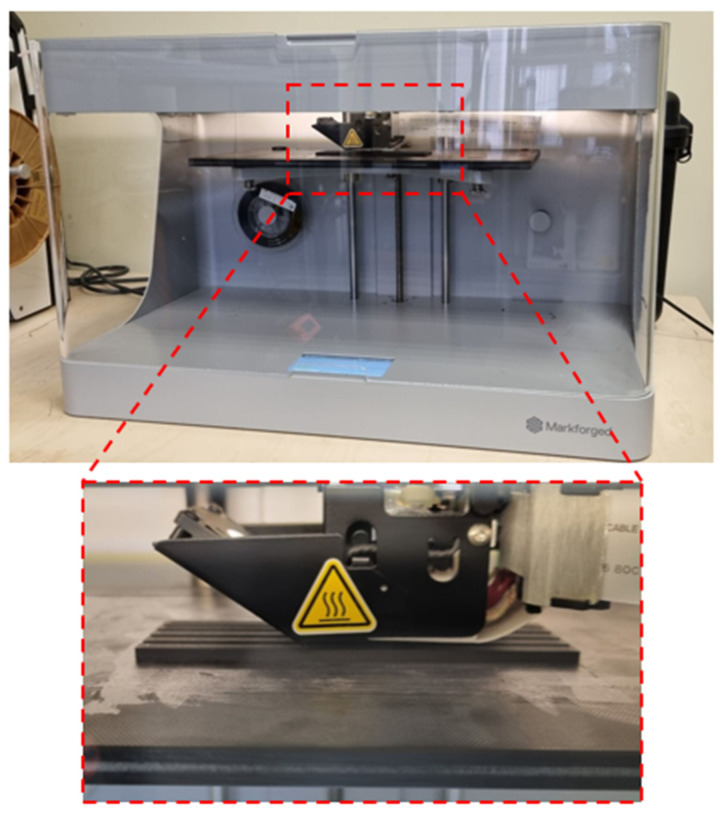
The Markforged Mark Two printer and a close-up of the print head during the process of manufacturing the test samples.

**Figure 7 materials-17-02937-f007:**
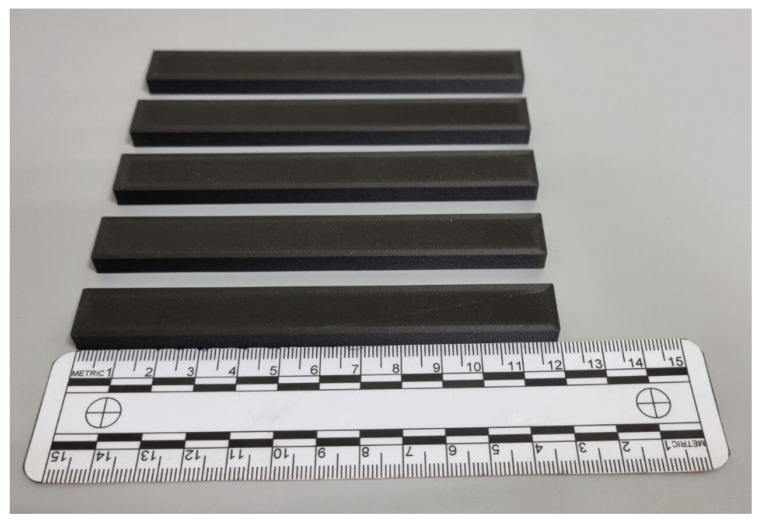
The samples were fabricated using the MEX (material extrusion) technique.

**Figure 8 materials-17-02937-f008:**
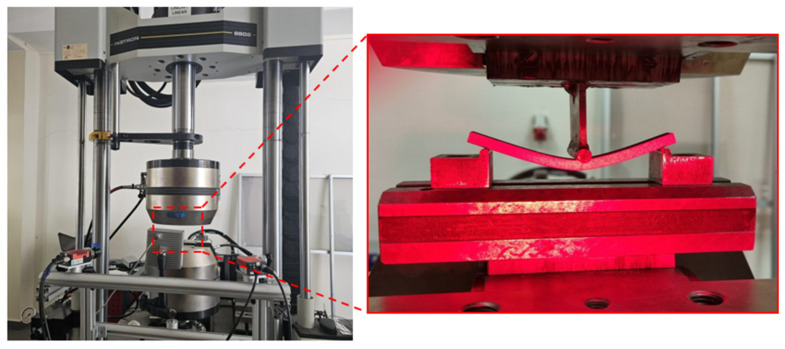
The Instron 8802 strength testing machine during a three-point bending test.

**Figure 9 materials-17-02937-f009:**
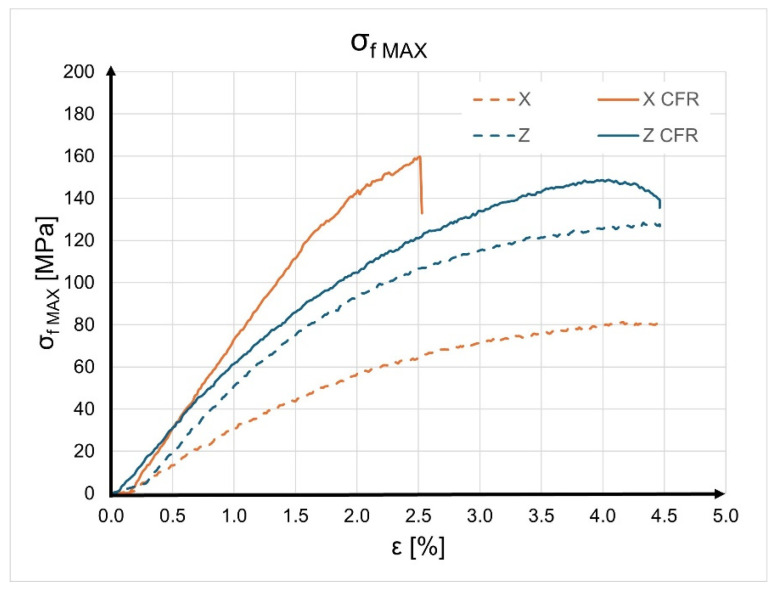
Curves of flexural stress (σ_f_) versus flexural strain (ε).

**Figure 10 materials-17-02937-f010:**
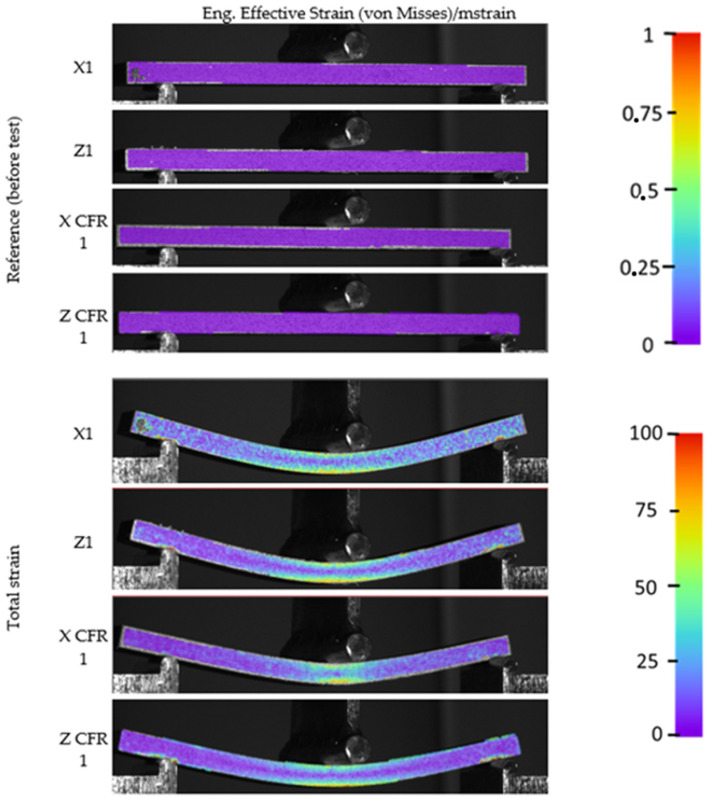
Strain levels registered via the DIC method.

**Figure 11 materials-17-02937-f011:**
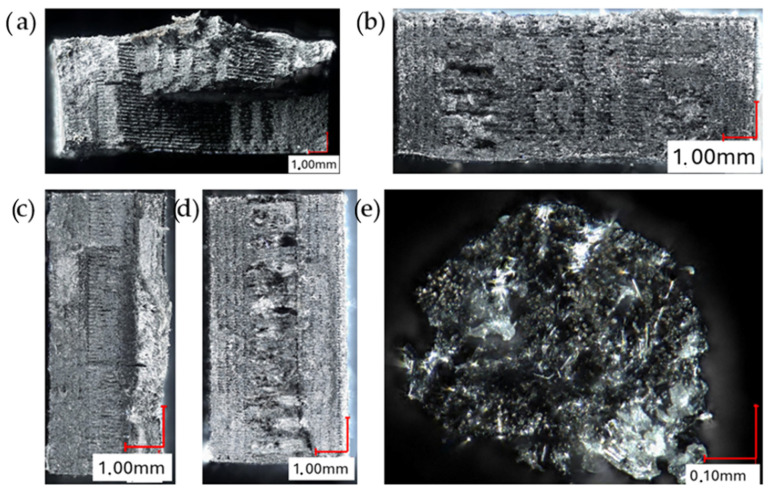
Fracture images of samples (**a**) “X”, (**b**) “X CFR”, (**c**) “Z”, (**d**) “Z CFR”, (**e**) continuous carbon fiber.

**Figure 12 materials-17-02937-f012:**
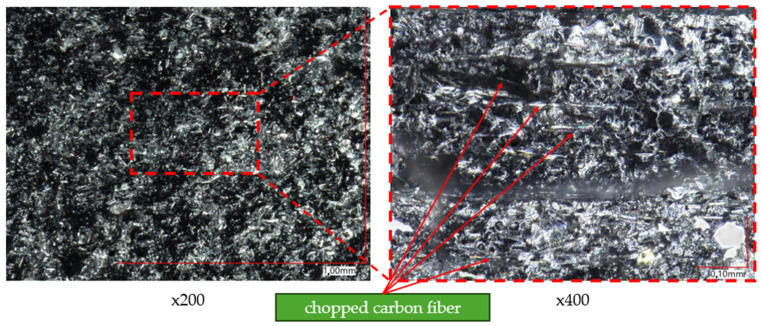
Onyx material without fiber.

**Figure 13 materials-17-02937-f013:**
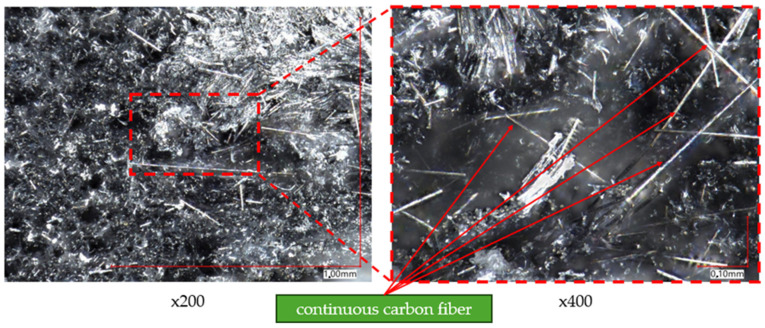
Onyx material with continuous carbon fiber.

**Figure 14 materials-17-02937-f014:**
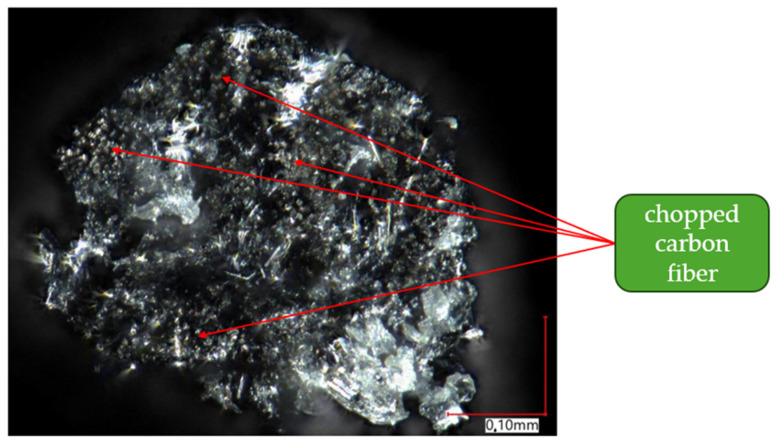
Cross-section of single continuous carbon fiber with indication of chopped carbon fiber particles.

**Table 1 materials-17-02937-t001:** Properties of used materials from Markforged company.

Property	Nylon	Onyx	Carbon Fiber
Density (g/cm^3^)	1.1	1.2	1.4
Tensile strength (MPa)	36	37	800
Tensile strain at break (%)	150	25	1.5
Tensile modulus (GPa)	1.7	2.4	60

**Table 2 materials-17-02937-t002:** Parameters used in MEX process.

Parameter	Value
Sample infill	100%
Bed heating	(unheated bed)
Nozzle temperature for Onyx filament	280 °C
Nozzle temperature for carbon fiber	260 °C
Layer thickness	0.125 mm
Number of outer perimeters	5

**Table 3 materials-17-02937-t003:** The values recorded and calculated based on the conducted bending tests.

Sample		Flexural Strength σ_fM_ [MPa]	Flexural Strain ε_fB_ [%]	Young’s Modulus E [MPa]
“X”	1	71.47	4.46	2789
	2	79.98	4.40	3639
	3	81.26	4.16	4022
	4	75.61	4.47	3362
	5	78.39	4.36	4113
“X CFR”	1	146.30	2.08	8852
	2	146.31	1.89	8646
	3	128.30	2.40	8040
	4	154.67	2.45	9317
	5	159.55	2.51	8851
“Z”	1	133.59	4.36	6868
	2	128.33	4.33	5095
	3	131.52	4.39	6372
	4	132.53	4.27	6933
	5	114.50	4.27	5733
“Z CFR”	1	146.58	3.85	6998
	2	148.73	4.04	6989
	3	136.14	4.20	6155
	4	143.77	4.02	6766
	5	141.65	3.64	6815

**Table 4 materials-17-02937-t004:** Average values and their deviations calculated based on the bending test results.

Samples	Average of Flexural Strength σ_fM_ [MPa]	Standard Deviation Δ σ_fM_ [MPa]	Average of Flexural Strain ε_fB_ [%]	Standard Deviation Δε_fB_ [%]	Young’s Modulus E [GPa]	Standard Deviation ΔE [GPa]
“X”	77.34	3.49	4.38	0.11	3.56	0.48
“X” CFR	147.03	10.64	2.27	0.24	8.74	0.41
“Z”	128.10	7.02	4.34	0.04	6.20	0.70
“Z” CFR	143.38	4.35	3.95	0.19	6.74	0.31

## Data Availability

Data are contained within the article.

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
