# Peer review of "Bending Strength of Continuous Fiber-Reinforced (CFR) Polyamide-Based Composite Additively Manufactured through Material Extrusion"

_materials, 2024, doi:10.3390/ma17122937_

Round 1

Reviewer 1 Report

Comments and Suggestions for Authors

In general, the paper shows interesting results and should be accepted for publication after minor revision. It is necessary to find out the following questions:

Why were carbon fibers chosen to make the polyamide-based composite, which was made by extrusion of the material? A discussion of the compatibility between the polyamide matrix and the carbon fibers should be included in the results.

What are the advantages and disadvantages of the studied technique and the composite obtained compared to a similar one?

What about the impact on the other parameters?

Comments on the Quality of English Language

Minor editing of English language required.

Author Response

In the beginning, we would like to thank you for your revision and comment which were very helpful in improving our work. Below you can find our answer to your comments. I am sending an updated version of the article as an attachment. The text has been modified from the previous version. The language correction has been marked in the change tracking mode. Modifications are highlighted in yellow.

  1. Why were carbon fibers chosen to make the polyamide-based composite, which was made by extrusion of the material? A discussion of the compatibility between the polyamide matrix and the carbon fibers should be included in the results.

Response 1:. On page 13, at the end of the Results chapter, the motivation for the use of carbon fibers and polyamide in the manufacture of composites is included.

  1. What are the advantages and disadvantages of the studied technique and the composite obtained compared to a similar one?
    Response 2: One point was added (point number 6) in the final conclusions in which the advantages and disadvantages of the MEX-CFR technique compared to the MEX technique are described
  2. What about the impact on the other parameters.
    Response 3: The conclusions in point 7 include the effect of the use of carbon fibers on other material properties.

Reviewer 2 Report

Comments and Suggestions for Authors

Please read the attachment. Thank you.

Author Response

In the beginning, we would like to thank you for your revision and comment which were very helpful in improving our work. Below you can find our answer to your comments. I am sending an updated version of the article as an attachment. The text has been modified from the previous version. Due to the removal and addition of some illustrations, the numbering has changed. The language correction has been marked in the change tracking mode. Modifications are highlighted in green.

  1. Keywords: Please provide between 5 and 10 keywords that should not repeat the words/phrases that appeared in the manuscript title.

Response 1: Keywords were changed.

  1. Introduction: Please introduce the main contributions of this study to the fields. Please clearly mention the gaps in this study. Please add a paragraph to introduce the outline of the manuscript.

    Response 2: A description of the contribution of the research data in the field of property analysis of composite materials is given at the end of the chapter ‘Introduction’
  2. Please add the flow chart for the study process.

    Response 3: The flow chart is shown in Figure 1.

  1. Figure 3 should be removed. I think it is too simple and general.

    Response 4: Figure 3 has been removed.

  1. Clarity and Detail in Methodology: The manuscript would benefit from a more detailed description of the specimen fabrication process, including specific parameters used during the MEX additive manufacturing. This would enhance reproducibility and provide a clearer understanding of the experimental setup.

    Response 5: A description of the manufacturing process is included between Figure 5 and Table 2. Table 2 contains the most important parameters of the MEX manufacturing process. In addition, Figure 4 shows the two orientations of the samples as they were fabricated. The specimen markings (along the X-axis and Z-axis) are described in the bullets.

  1. To improve the literature review, please read the following works to better understand the recent development of AM techniques. (i) WAAM Technique: Process Parameters Affecting the Mechanical Properties and Microstructures of Low-Carbon Steel, (ii) Trajectory Strategy Effects on the Material Characteristics in the WAAM Technique

    Response 6: Together with all the co-authors, I would like to express our gratitude for submitting these highly valuable articles. They represent a significant step forward in the development of metal additive manufacturing techniques. Although they have a limited connection to the main topic of our article, we are conducting a detailed analysis of the submitted works. The given articles have been quoted in this manuscript

  1. The conclusion could be expanded to discuss potential findings' potential applications and suggest future research directions. This would provide a broader perspective on the implications of the study.

    Response 7: In the conclusions, an eighth point has been added which describes the applicability of the results of the research in question and the directions for their development.

  1. How did you validate the results?

    Response 8: To validate the results, we conducted a series of five tests on samples produced using the MEX-CFR technique. The obtained data were compared with results from control samples made using the standard MEX technique. Additionally, a microstructural analysis was performed to ensure that the fibers were properly bonded within the matrix.

  1. What are the main limitations of this approach?

    Response 9: The main limitation of the MEX-CFR technique is the increased complexity and cost associated with processing carbon fiber-reinforced composites. The abrasive nature of carbon fibers leads to faster wear and tear on the extrusion nozzles and other components, resulting in shorter tool lifespans. Additionally, the precise control required during the manufacturing process can make it challenging to achieve consistent quality and performance. The higher material costs of carbon fibers compared to standard polymers further add to the overall expense. Moreover, potential issues with fiber alignment and distribution within the matrix can affect the uniformity of the material properties.

    This information was shortly described in revised version of the manuscript (point 6 in conclusions).

  1. How does the addition of continuous carbon fiber reinforcement impact the mechanical properties of other polymer composites produced through Material Extrusion (MEX) additive manufacturing?

    Response 10: In the revised version of the manuscript, points 6-7 in the conclusions section describe the advantages and disadvantages of using carbon fiber as reinforcement. The data regarding the impact of carbon fiber on the material's strength are included in point 2 of the conclusions.

  1. What are the potential industrial applications for polyamide composites reinforced with continuous carbon fiber, and how do these applications benefit from the enhanced mechanical properties observed in this study?

    Response 11: Point 8 in the conclusions describes the potential applications of the given manufacturing technique.